# Visualized in-sensor computing

Yao Ni [1,2,3], Jiaqi Liu[1,2,3], Hong Han [1,2,3], Qianbo Yu[1,2], Lu Yang[1,2], Zhipeng Xu[1,2], Chengpeng Jiang [1,2], Lu Liu[1,2] & Wentao Xu [1,2] ✉

In artificial nervous systems, conductivity changes indicate synaptic weight updates, but they provide limited information compared to living organisms. We present the pioneering design and production of an electrochromic neuromorphic transistor employing color updates to represent synaptic weight for in-sensor computing. Here, we engineer a specialized mechanism for adaptively regulating ion doping through an ion-exchange membrane, enabling precise control over color-coded synaptic weight, an unprecedented achievement. The electrochromic neuromorphic transistor not only enhances electrochromatic capabilities for hardware coding but also establishes a visualized pattern-recognition network. Integrating the electrochromic neuromorphic transistor with an artificial whisker, we simulate a bionic reflex system inspired by the longicorn beetle, achieving real-time visualization of signal flow within the reflex arc in response to environmental stimuli. This research holds promise in extending the biomimetic coding paradigm and advancing the development of bio-hybrid interfaces, particularly in incorporating color-based expressions.

Artificial neural systems leveraging ion conduction play a crucial role in advancing universal artificial intelligence, neural robotics, in vitro perception of motion prostheses, and the replacement of diseased nerves[1-4]. Within such architectures, the reconfiguration of synaptic weight stands as a key function, enabling the autonomous encoding of spatiotemporal information in response to stimuli. This capability empowers artificial neural reflex systems to adapt and respond to a dynamic external environment[1-4]. However, in most artificial nervous systems, changes in conductivity serve as the primary indication of synaptic weight updates, offering limited information compared to the complexities observed in living organisms[5-7]. The uncertainty and intricacy of ion doping and de-doping during action potentials result in ambiguous relaxation times for weight updates, potentially leading to challenges in processing time-series signals, such as aliasing or misses[8-10]. Exploring alternative forms of synaptic weight information could mitigate these issues.

The communication of color information stands as one of the most immediate and widespread methods of interaction among biological entities[11]; it indicates that the ability to integrate color information into artificial neural systems can offer numerous advantages

and opportunities for enhancing their functionality. For instance, chameleons can adjust the activity of pigment cells in response to their surroundings by modulating the concentration of catecholamine neurotransmitters like epinephrine (EPI), enabling rapid color changes for communication with conspecific individuals (Fig. 1a)[12]. Despite the direct and efficient nature of color weight updates for information exchange, there has been a notable scarcity of neuromorphic electronics capable of utilizing color changes to convey information.

In most organisms, the neuro-reflex process adheres to an all-or-none law, where an action is triggered only when the stimulus intensity surpasses a certain threshold[13-15]. Most existing neuromorphic electronics focus primarily on pure electrical signal processing and do not fully exploit the potential of incorporating color-based information; such reflex systems rely solely on the electrical signal transmission and might inadvertently filter out certain weak yet potentially significant signals[16-18]. By combining color-based alterations with adaptable electrical properties, it becomes possible to visualize changes in synaptic weight and monitor a broader range of environmental signals compared to conventional bionic reflex systems. Moreover, the integration of color information as an additional measure of synaptic weight in

[1]Institute of Optoelectronic Thin Film Devices and Technology, Key Laboratory of Optoelectronic Thin Film Devices and Technology of Tianjin, College of Electronic Information and Optical Engineering, National Institute for Advanced Materials, Nankai University, Tianjin 300350, China. [2]Shenzhen Research Institute of Nankai University, Shenzhen 518000, China. [3]These authors contributed equally: Yao Ni, Jiaqi Liu, Hong Han. ✉e-mail: wentao@nankai.edu.cn

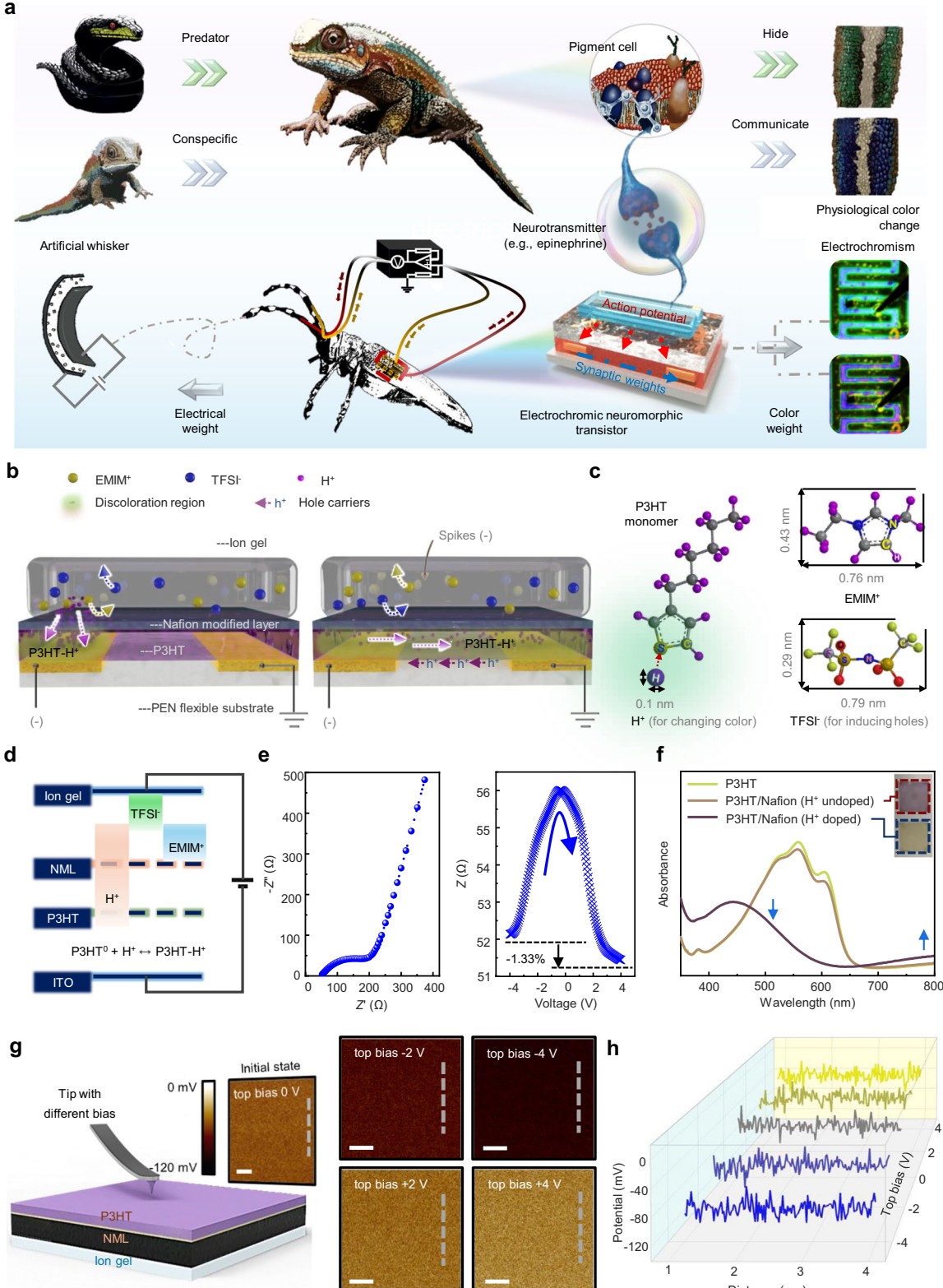

**Fig. 1 | Design of an electrochromic neuromorphic transistor (ENT).**
**a** Applications of ENT in visualized in-sensor computing analogous to physiological color change responses by a chameleon and implementing a visualized bionic reflex system. **b** Schematic diagram and working mechanism of an ENT device. **c** Chemical structures of P3HT monomer, H⁺, EMIM⁺, and TFSI⁻. **d** Distribution of ions for a diode device of architecture glass/ITO/P3HT/NML/ion gel, under a condition of applying electric pulse. **e** EIS analysis for a diode device of architecture glass/ITO/P3HT/NML/ion gel. Under electrochemical scanning, resistance changes for the diode device. **f** UV–Vis absorbance spectra as a function of wavelength for pristine P3HT, P3HT/Nafion (H⁺ undoped), and P3HT/Nafion (H⁺ doped) films. Insets: optical images of the change in color of the bilayer film. **g** Corresponding CPD of a diode device of architecture glass/ITO/ion gel/NML/P3HT using the KPFM mode after top bias of 0 (initial state), −2, −4, 2 and 4 V on the AFM tip. Scale bars, 1 μm. **h** CPD values extracted from the selected data line in (**g**).

visualized artificial neural systems can enhance the accuracy and efficiency of in-sensor computing tasks by high-fidelity processing data locally[19–21]. This approach enables more nuanced processing capabilities, mirroring the complexity and richness observed in natural sensory systems.

Here, we present a conceptual design and production of the inaugural ENT, which incorporates color weight updates alongside electrical weight. This innovation harnesses the electrochromic effect generated by proton doping within a crystallized polymer, as well as an anion accumulation induction effect on the charge. The utilization of a Nafion film as an ion-exchange membrane allows adaptive control of doping by different ions, facilitating the precise adjustment of two synaptic weights: chromaticity and conductivity. Notably, our device boasts a swift reset time, with conductivity reverting to 1/3000th of its initial value in just around 1 s, while chromaticity concurrently resets to 0. This rapid recovery time ensures unhindered, rapid signal transmission. Leveraging these attributes, our designed ENT showcased diverse bionic coding applications, including coding via combined conductivity and chromaticity enhancement, and the establishment of a visualized pattern-recognition network. Integration of the ENT with an artificial whisker, capable of both external vibration signal perception and feedback actuation, enables the creation of a bionic reflex system. Notably, this system marks an instance of utilizing color weight changes to visually represent signal flow within the reflex arc in response to environmental stimuli.

## Results

### Design of electrochromic neuromorphic transistor

Drawing inspiration from creatures like the chameleon and longicorn, capable of adapting their skin tones by adjusting nanocrystal contractions in pigment cells to match environmental changes or communicate, we designed an artificial nerve featuring an electrochromic function for a visualized neural reflex system (Fig. 1a). At the core of this biomimetic system is the ENT, serving as a pivotal component. The ENT emulates the release mechanism of biological neurotransmitters (e.g., EPI) in response to diverse environmental stimuli captured by artificial whiskers. It operates as both a motor nerve-to-muscle junction, directing commands to the artificial whiskers for inducing feedback contractions, and as an encoder capable of assigning tunable color weight to offer an alternative means of expressing information.

An ENT device is composed of an ion gel, a Nafion modified layer (NML), a poly(3-hexylthiophene) (P3HT) channel and two metal contact pads as source/drain (S/D) electrodes on a polyethylene naphthalate (PEN) flexible substrate (Fig. 1b). The ion gel primarily comprises 1-ethyl-3-methylimidazolium (EMIM$^+$) cations, bis(trifluoromethylsulfonyl)imide (TFSI$^-$) anions, and hydrogen (H$^+$) protons (Fig. 1c). Functionally, the ENT device mirrors the neurotransmitter release mechanism between neurons: Upon application of presynaptic spikes to the ion gel, mobile ions migrate within the electrical field. Accumulation of TFSI$^-$ anions at the interface of the ion gel/NML triggers hole carriers in the P3HT channel, akin to neurotransmitters in the synaptic cleft, eliciting an excitatory postsynaptic current (EPSC) in its dendrite[22]. Furthermore, the introduction of H$^+$ into P3HT channels via NML initiates an electrochromic process resembling the action of biological neurotransmitters in regulating pigment cells for physiological color changes[12].

We engineered a crystallized P3HT nanowire (NW) thin film through a low-temperature solvent engineering technique, showcasing its remarkable capacity for synaptic weight modulation in artificial synapse transistors[23]. Atomic force microscopy (AFM) details the distinct and well-defined P3HT NW structures, forming a 12 nm interconnected network that efficiently transports charge carriers (Supplementary Figs. 1a and 2a). Both the P3HT layer and the upper NML exhibit smooth surfaces critical for establishing seamless contact with the ion gel (Supplementary Fig. 1 and Fig. 2b). Scanning electron

microscopy (SEM) imaging verifies this surface quality (Supplementary Fig. 3).

High-resolution transmission electron microscopy (HRTEM) confirms the creation of the P3HT NW thin film through self-assembly of polymer chains (Supplementary Fig. 4a). Notably, the HRTEM display of the Nafion film presents lattice fringes, suggesting its capability for ion-exchange purposes, selectively filtering ions within the ion gel (Supplementary Fig. 4b). Functioning as a proton-exchange membrane, Nafion integrates H$^+$ into the P3HT backbone upon the application of an electric pulse (Fig. 1d)[24].

Moreover, Nafion serves as a Brønsted solid superacidic electrode binder, aiding P3HT in capturing additional H$^+$ and generating enhanced channel conductivity through the production of delocalized positive charges (Supplementary Fig. 5, detailed in Supplementary Note 1)[25]. Our diode device, under electrochemical impedance spectroscopy (EIS) analysis, demonstrates commendable ion–transport efficiency and notable reductions in electric double-layer resistance following the application of a positive voltage (Fig. 1e). These outcomes underscore the promising functionality and efficiency of our developed system.

The introduction of H$^+$ into P3HT often accompanies an intriguing electrochromic phenomenon[25]. To investigate the nature of polythiophene doping at the P3HT/Nafion interface, we examined the ultraviolet-visible (UV–Vis) spectra of pristine P3HT, P3HT/Nafion (H$^+$ undoped), and P3HT/Nafion (H$^+$ doped) films (Fig. 1f). The UV–Vis spectra reveals a broad peak at 780 nm induced by H$^+$ doping in the P3HT/Nafion, absent in the pure P3HT or undoped bilayer films. Concurrently, as this new peak emerges, the primary P3HT absorbance peak at 540 nm diminishes, causing a visible color change in the P3HT/Nafion bilayer films (Fig. 1f, insets). These observations are consistent with the optical signatures of bipolaron and polaron species within an H$^+$-doped P3HT-polymer backbone[26]. X-ray photoelectron spectroscopy (XPS) corroborates evidence of intermixing of the two species upon H$^+$ doping (Supplementary Fig. 6, detailed in Supplementary Note 2).

For a deeper understanding of the ion accumulation in response to bias, we investigated the contact potential difference distribution (CPD) of a P3HT film, engineered based on an ion gel/NML architecture, using in situ Kelvin probe force microscopy (KPFM) after applying various biases to the top terminal (Fig. 1g). In its initial state, the average surface potential ($p_{as}$) is measured at −53.3 mV. Subsequent observations reveal lower $p_{as}$ values of −75.2 and −92.2 mV under top biases of −2 and −4 V, respectively, confirming H$^+$ doping in the P3HT film. Conversely, with top biases of 2 and 4 V, $p_{as}$ values increase to −43.7 and −37.8 mV, signifying the H$^+$ de-doping process. Figure 1h provides detailed CPD values extracted from the selected data line.

### Electro-chromaticity boost coding

The ENT device can remarkably achieve both an EPSC and an electrochromic response upon external electrical stimulation. Under an external electric field, the ions within the ion gel undergo movement, altering the device state (Fig. 2a, b). At a no-bias voltage condition, both anions and cations in the ion gel are randomly dispersed (state #1). With a driving voltage of −0.5 V applied to the drain terminal, cations (EMIM$^+$ and H$^+$) migrate toward the drain electrode; however, only H$^+$ could pass through the NML and enter the P3HT channel. This process coincides with a visible electrochromic alteration in the drain region (state #2). When a brief presynaptic spike of −4 V is applied to the gate terminal, it leads to anion accumulation (TFSI$^-$) at the ion gel/NML interface, attracting additional holes in the conductive channel to form EPSC. Simultaneously, H$^+$ ions diffuse from the drain electrode across the interface, inducing a transient spread of electrochromic effects across the entire interface (state #3). Following the spike, ion distribution gradually returns to state #2, and the EPSC rapidly decays,

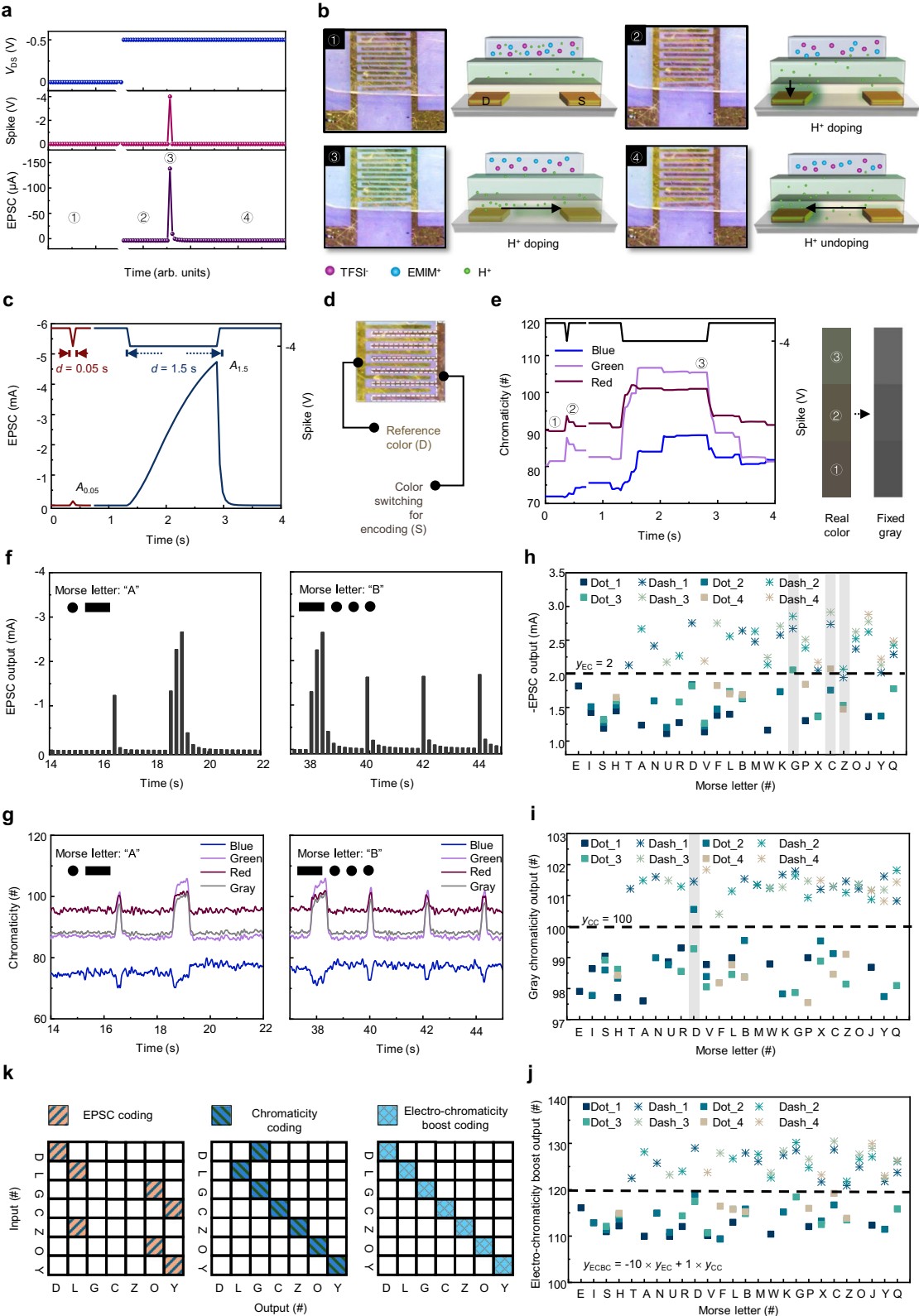

with the electrochromic effect vanishing except in the drain area (state #4).

In biological synapses, short-term plasticity (STP) denotes a temporal modulation of synaptic strength, swiftly reverting the post-synaptic current to its initial state within hundreds to thousands of milliseconds[27]. By applying spikes with different durations ($d$) to the ENT device, it can be observed that spikes of 1.5 s induced a

more pronounced time-dependent accumulation of TFSI⁻ at the ion gel/NML interface, resulting in a substantial increase in the number of hole carriers (Fig. 2c). This phenomenon can mimic spike-duration-dependent-plasticity (SDDP), showing that the intensity of this response increases nearly linearly with $d$, following the SDDP index calculation $A_d/A_{0.05} \times 100\%$, where $A_i$ is the EPSC value under $d = i$ [s].

**Fig. 2 | Electro-chromaticity boost coding. a** EPSC in response to a −4 V input spike was read with a −0.5 V $V_{DS}$. **b** Top-view optical micrographs at four different device states ①–④ with different device operation history as shown in (**a**). The comb-like region is the Au electrode, and the electrochromic feature is due to $H^+$ ion overflow in the P3HT layer. Side view device sketches showing the distribution of ions at four different device states ①–④ as shown in (**a**). Arrows in the sketches indicate the drift/diffusion directions of $H^+$ ions. **c** EPSCs triggered by two spikes with durations of 0.05 and 1.5 s. **d** Fixed sampling area on the source electrode for analyzing color change. **e** Chromaticity of three primary colors extracted from nine sampling points triggered by two spikes with durations of 0.05 and 1.5 s; Chromaticity changes of real color and fixed gray corresponding to different triggering events. **f** Under electric spikes, EPSC outputs by the ENT representing the International Morse code of letters "A" and "B". **g** Under electric spikes, chromaticity outputs via ENT representing the International Morse code of letters "A" and "B". **h–j** The EPSC output, gray chromaticity output distribution, and electro-chromaticity boost output distribution of 26 Morse letters; the incorrectly coded letters marked with a gray rectangle background. **k** Confusion matrix of extraction results according to EC, CC, and ECBC modes.

Furthermore, the source region can be chosen as the color encoding area, evaluating the average chromaticity of the three primary colors in response to stimuli (Fig. 2d). Corresponding to the EPSC trend, the color alteration notably increases with longer $d$ durations; notably, the change in green chromaticity stood out among the three primary colors (Fig. 2e). Leveraging a colorimetric card depicting actual color or fixed gray, the current state of the ENT can be easily identified through visual inspection. The correlation between gray gain (ΔGray) and $d$ is notably high within 0.4 s, affirming its potential application in information coding. Comparing the decay of EPSC gain (ΔEPSC) and ΔGray value against the reset time, after a 1.5-s spike, ΔEPSC declines to 1/3000 of its initial value in just ~1 s, concurrently with ΔGray returning to 0. This is because Nafion hinders the doping of anions into the P3HT channel. Consequently, the anions swiftly return to their initial positions after the electrical spikes are removed, leading to the rapid restoration of the system to its initial state (Supplementary Fig. 7). This represents the initial neural morphological device exhibiting absolute STP in both electrical and color change responses, underscoring its high fault tolerance in real-time multimode signal processing.

The ENT can be effectively responsive to electrical impulses mimicking international Morse code, displaying promise for multimode signal coding applications. When activated by encoded short dot signals (−4 V, 0.2 s) and extended dash signals (−4 V, 0.8 s), the ENT swiftly generates various EPSC amplitudes corresponding to different English letters (Fig. 2f; Supplementary Fig. 8). Moreover, owing to its distinctive electrochromism, the ENT successfully validates received Morse signals by modulating color changes of different durations, marking a pioneering demonstration of this capability (Fig. 2g; Supplementary Fig. 9, Supplementary Movies 1 and 2).

In establishing coding mechanisms, a category score threshold of $y_{EC}$ in EPSC coding (EC) mode was set at 2; for chromaticity coding (CC) mode, the threshold $y_{CC}$ is designated as 100. Values below the threshold are classified as dot signals, while those surpassing the threshold are labeled as dash signals. The accuracy rates in EC and CC modes are 88.5% and 96.2%, respectively (Fig. 2h, i). Combining the outputs of both modes in a weighted manner introduces a coding method, electro-chromaticity boost coding (ECBC), with a threshold $y_{ECBC}$ of 120. This fusion allows each output signal to accurately map to an exclusive input signal, enabling the recognition of all letters with a 100% accuracy rate (Fig. 2j, k; Supplementary Table 1).

## Visualized pattern-recognition network

In biology, synaptic weight updates rely on the frequency of stimuli[28]. Similarly, in the ENT, as the spike number ($n$) increases, the induced hole carriers in the P3HT channel increase, leading to a proportional rise in EPSC (Fig. 3a). Parallel experiments affirmed a significant and reliable increase in the spike-number-dependent plasticity (SNDP) index as $n$ increased (Supplementary Figs. 10 and 11). The chromaticity change of the ENT similarly reflects the alteration in synaptic weight under different spike numbers (Fig. 3b). The ENT's short reset time of ~1 s makes it highly suitable for a visualized pattern-recognition network without the need for actively executing additional weight-refreshing actions (Supplementary Fig. 12): a neural core with a high refresh rate enables accelerated matrix multiplication achieved by

applying various spike number-voltage ($V_{DS}$) amplitudes (−0.1 to −0.5 V) and different spike numbers (1–100) (Fig. 3c).

In the domain of biological vision, visual information undergoes initial processing in the retina before transmission through visual pathways to the cerebral cortex, enabling perception. Within this complex system, visual memory plays a pivotal role by facilitating rapid information refreshing while retaining crucial signal processing capabilities. This preprocessing of signals at the perceptual end enhances computational efficiency and reduces energy consumption in the central brain. Drawing inspiration from the efficacy and transient nature of visual memory, we designed a 3 × 3 ENT array to mimic the visual cortex (Fig. 3d). Each ENT functions within a crossbar array, operating in a voltage-controlled capacity in the circuit schematic, performing matrix multiplication and outer-product update operations: $V_{DS}$ serves as the reading voltage controlled by the extracted eigenvectors from classified samples simulating the reception of visual information. The conductivity state is adjusted by varying the number of pulses applied to the device within a short period, assigning different weights to the received visual information; the current programming is determined by the product of the reading voltage signal and conductance state, executed through the crossbar to output preprocessed results.

To demonstrate recognition capability, four types of images (X, L, T, Y) were input into the neural network (Fig. 3e). The nine-channel input layer, represented by different $V_{DS}$, reflects pixel information. For each channel, numerical weights are mapped onto the conductance states of the ENT devices, regulated by varying spike numbers applied to the devices. This enables the multiplication input of the device to generate a current peak, which, when summed across all channels, is utilized in neuron calculation. Using X as an example, the optimal number of spikes applied to each ENT can be obtained through training; subsequently, the weighted peak current output from the nine channels can be mapped to the classification label (Fig. 3f). With an increasing maximum number of pulses ($n_{max}$), the optimal spike count applied to each input neuron is continuously adjusted, broadening the output value range for each image category, resulting in improved recognition accuracy (Fig. 3g; Supplementary Fig. 13). Additionally, real-time changes in synaptic weights can be visualized during the training process (Fig. 3h; Supplementary Fig. 14): we can retrieve the EPSC mapping value by referring to a colorimetric card that holds the chromaticity information of red-green-blue (RGB) and fixed gray.

A brain-inspired learning architecture can adapt the neural network's structure in response to the input distribution, enabling efficient resource management (Supplementary Fig. 15)[29]. Exploiting the temporal correlation plasticity of ENT devices and their runtime reconfigurability, a biologically inspired time-series neural network (TSNN) architecture was developed, showcasing a self-adaptive dynamic grow-when-required (GWR) characteristic (Supplementary Figs. 16–19, detailed in Supplementary Note 3).

Moreover, synapses with a low probability of neurotransmitter release exhibit a low-frequency suppression filtering effect: allowing the passage of signals with frequencies exceeding a cut-off value ($f_c$), while suppressing signals below this threshold[30,31]. Our ENT exhibits similar low-frequency suppression filtering behavior, responding to

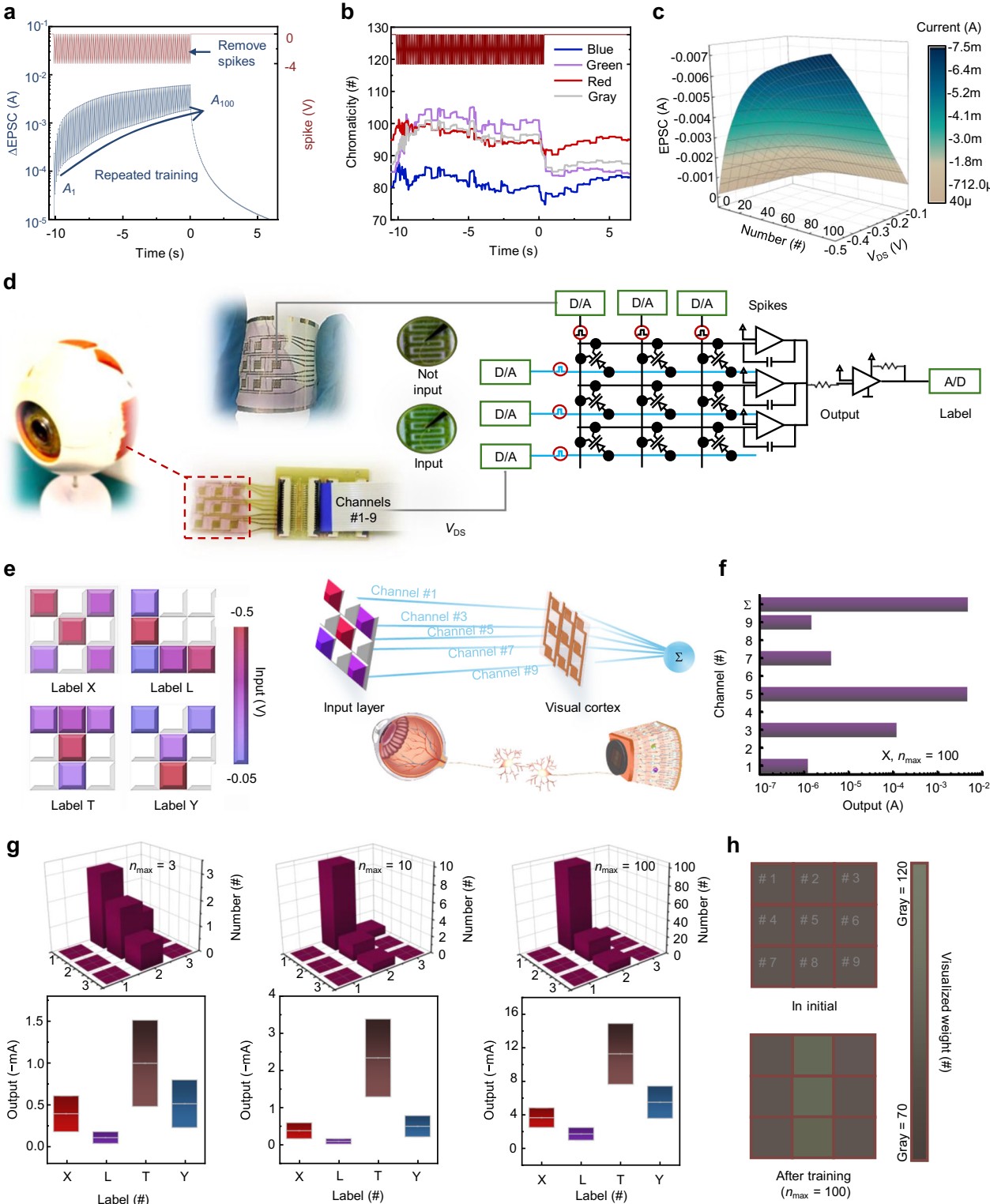

**Fig. 3 | A visualized pattern-recognition network. a** ΔEPSC triggered by 100 spikes of −4 V. **b** Chromaticity of three primary colors and gray extracted from nine sampling points triggered by 100 spikes of −4 V. **c** Current (EPSC)-spike number-voltage ($V_{DS}$) contour map of the ENT device. **d** Diagram of a 3 × 3 ENT array for visualized visual cortex. **e** Classification of 3 × 3-pixel images of X, L, T and Y, where the pixels are read by $V_{DS}$ of input neurons and processed by pulses from biomimetic visual cortex. **f** Schematic of the response of each channel output and total output in an example of image X, under $n_{max} = 100$. **g** With different $n_{max}$, the optimal number of pulses applied to each input neuron, and the range of output value for each image category. Box plots indicate standard deviations of output value. **h** Visualized weight changes in initial and after training with $n_{max} = 100$.

stimuli with varying frequencies, exemplified by inputting a city image to simulate dynamic filtering functions (Supplementary Figs. 20–23, detailed in Supplementary Methods and Supplementary Note 4). These patterns could encode large-scale, real-time dynamic signals.

## Visualized artificial neural reflex arc

The longicorn beetle boasts a color-changing reflex system, wherein pigment cells on its body surface adapt their color based on the intensity and frequency of external stimuli, facilitating environmental adaptation and concealment. An analogous visualized artificial neural reflex system can be devised to respond realistically to external stimuli. This system not only autonomously adjusts its colors to blend in with subtle environmental changes for camouflage but also assigns color change sequences to specific information for communication, ensuring secure encoded transmission.

By introducing color weight updates as an alternative measure of plasticity, our ENT achieves the visualization of synaptic weight changes. As a proof of concept, we connected the ENT to an artificial whisker through bionic afferent and efferent nerves, forming a neuro-electronic reflex arc akin to that of the longicorn beetle (Fig. 4a, Supplementary Figs. 24 and 25). This artificial whisker consists of a vibration-sensing layer and a bionic actuation layer, enabling it to sense external vibration signals and execute actions. The ENT serves as a nerve center, transforming stimulus signals from the artificial whisker into control instructions and providing feedback to the artificial whisker for action responses. Notably, the ENT also functions as a synaptic weight indicator, changing color and blinking in real time in sync with the processed information flow.

To assess the flexibility of the ENT during biointegration, a nominal stress–strain test was conducted (Supplementary Fig. 26). As the tensile strain increased, the ENT exhibited successive stages, including the elastic stage, plastic yield stage, and necking stage before the eventual fracture. At the point of fracture, the elongation reached 73.77%. The ENT demonstrates a maximum force tolerance of 391.72 N and a maximum tensile strength of 217.62 MPa, meeting the necessary criteria for biointegration.

In the artificial whisker, a core component is a carbon nanotube/poly-dimethylsiloxane (CNT/PDMS) composite film that serves as the vibration sensor (Fig. 4b, detailed description in Supplementary Methods). When external forces are applied to the composite film, conductive pathways are enhanced, allowing the transmission of a greater current. By continually contacting a target object with the artificial whisker, regular pulse fluctuations can be monitored, indicating different vibration states: as the vibration amplitude of the target object transitions from weak to strong, the pulse duration and period gradually increase.

For bionic actuation, we incorporated an ionic polymer–metal composite into the artificial whisker (Fig. 4c, detailed description in Supplementary Methods)[32]. The creation of hydrated sodium ions is pivotal during the alkaline washing step in the fabrication process to ensure the functionality of the artificial whisker. Specifically, within the main body of the artificial whisker, the alkaline washing step induces the formation of hydrated sodium ions. When subjected to an electric field, the whisker's end undergoes expansion and deflection towards the anode, propelled by the migration of hydrated sodium ions from the anode to the cathode.

We demonstrated the flow of chromaticity signals and electric signals processed by this visualized bionic reflex system under external environmental stimulation (Fig. 4d). A series of vibration stimuli of varying intensities are received by the artificial whisker, subsequently converted into sampling spikes by the afferent nerve module and transmitted to the ENT. The spike sequences from V1 to V5 correspond to vibration intensities from weak to strong, and the intensities of the chromaticity (RGB and gray) and the output EPSC also increase correspondingly.

During this process, the chromaticity changes and blinking of the ENT, as well as the deflection angle of the artificial whisker, were recorded. In line with the all-or-none law, EPSCs triggered by V1 and V2 are insufficient to reach the threshold for eliciting an action potential, thus failing to drive the artificial whisker. Only under the application of stronger stimuli (i.e., V3–V5), does EPSC increases beyond the threshold level to induce a deflection in the whisker (Fig. 4e).

Furthermore, the ENT exhibits high-frequency color changes in response to spike sequences, even under the weakest V1 spikes. As the duration of stimulation increases, the significance of the chromaticity change weight significantly improves (Fig. 4f, detailed description in Supplementary Note 5). This blinking process provides a visual means to monitor seemingly insignificant yet vital information during action responses. The integration of motor responses with color information in this concept contributes to proposing paradigms for artificial neural systems that leverage ion conduction (Supplementary Table 2)[14,18,33–38].

## Discussion

We introduced an ENT designed for visualized in-sensor computing. This neuromorphic device represents a significant advancement to utilize the update of chameleonic color changes as a measure of synaptic weight. It showcases visualized weight updating, simulating the multidimensional tuning of synaptic weight by adjusting both neural impulse information weight and pigment cell chromaticity information weight. Leveraging a mechanism that adaptively screens ions through the NML, the device also achieves rapid reconfigurable electrical characteristics. Notably, the process exhibits a remarkably swift recovery time of less than 1 s, facilitating rapid signal transmission without interference. Capitalizing on these inherent features, we demonstrated an ECBC approach and developed a strategy for constructing a visualized pattern-recognition network. Moreover, the integration of the ENT with an artificial whisker led to the creation of a visualized neural reflex mirroring that of the longicorn beetle. This achievement marks the visualization of signal flow, effectively enabling the monitoring of minute environmental stimuli during reflex actuation. Our work has the potential to enhance the functionality and adaptability of artificial neural systems, thereby paving the way for the development of more sophisticated and bioinspired computing systems.

## Methods

The Supplementary Information displays more experimental details.

### Material preparation

P3HT solution was prepared by adding dichloromethane (DCM, Purchased from Macklin) to well-dissolved P3HT (Purchased from Sigma-Aldrich) in chlorobenzene (CB, Purchased from Macklin) (2 mg mL$^{-1}$) with CB: DCM volume ratio = 1: 1. The solution was heated at 60 °C for 2 h, then cooled slowly to room temperature to form crystallized NWs. Nafion (Purchased from DuPont) (0.5 wt%) was dispersed in a mixture of water and ethanol with a volume ratio = 1:1. Ion gels were prepared in ambient by dissolving poly(vinylidene fluoride-co-hexafluoropropylene) (PVDF-HFP, Purchased from Sigma-Aldrich) and the ionic liquid EMIM-TFSI (Purchased from Innochem) in a mass ratio of 1: 3 in acetone. The resulting ion-gel solution was stirred at 40 °C for at least 30 min, then drop-cast onto a prepatterned mold, then allowed to set in ambient air at room temperature to form a film.

### Device fabrication

The ENT device was fabricated in a top-gate bottom-contact structure. Firstly, source and drain electrodes of gold (60-nm layers) were prepared on a PEN (Purchased from Xiangcheng Technology) flexible substrate by thermal evaporation. Then, the substrate was preheated at 120 °C for 10 min. Subsequently, the P3HT solution was spin-coated at 2000 rpm for 30 s onto the source/drain electrodes and then

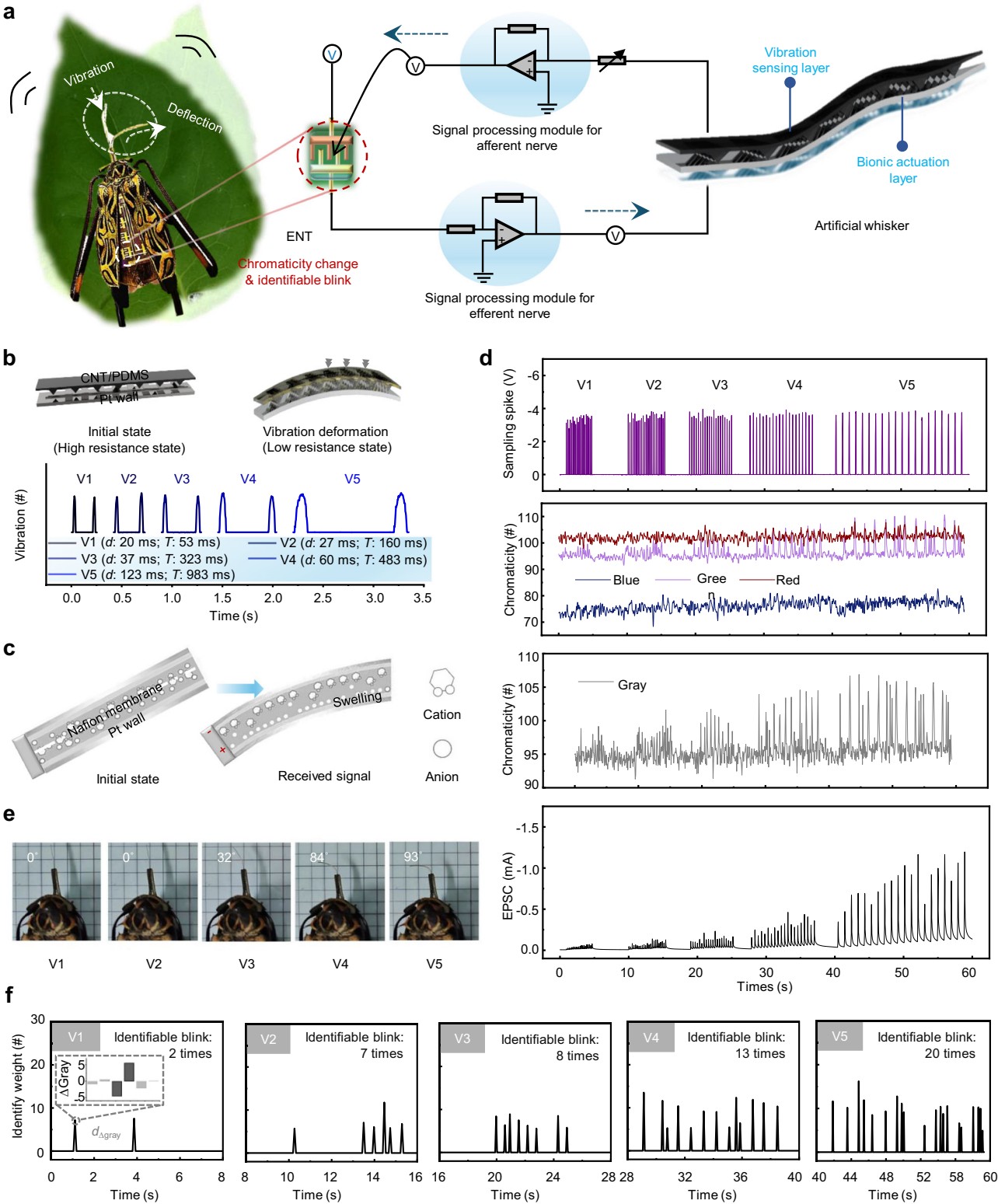

**Fig. 4 | A visualized bionic reflex system. a** A longicorn neuro-electronic system: ENT as neural signal processing center integrated into the body of the longicorn beetle, signal processing modules for afferent and efferent nerves, and artificial whisker combining both functions of sensing external vibration signals and realizing actuation. **b** Structure and motion mechanism of vibration sensing layer. **c** Structure and motion mechanism of bionic actuation layer. **d** Sampling spikes, chromaticity (RGB and gray) and the output EPSC of ENT when the artificial whisker senses different degrees of vibration (from V1 to V5: from weak to strong). **e** Deflection angle of the artificial whisker under different degrees of vibration. **f** Chromaticity change and identifiable blink of ENT under different degrees of vibration; the illustration shows the gray variation of an effective identified weight.

annealed for 10 min at 120 °C to crystallize to NW thin film. The Nafion dispersion was spin-coated onto the P3HT film at 2000 rpm for 30 s. Finally, ion-gel top-gate dielectric film was transferred onto the channel area.

### Device characterization

AFM images were obtained using a Bruker dimension icon microscope in tapping mode. To investigate the surface potential, the Bruker AFM system was switched in situ to a KPFM mode using various tip voltages (SCM-PIT-V2; materials: 0.01–0.025 Ω cm Antimony (n)-doped Si; frequency: 75 kHz; spring constant: 3 N m$^{-1}$; diameter of tip: 50 nm), at a scan rate of 1 Hz with a resolution of 256 pixels × 256 pixels. SEM images were performed using an FEI-Apreo field emission microscope. HRTEM observations were obtained using a JEM-2100F microscope. The optical absorption spectra were obtained using a UV–Vis spectrophotometer (Cary 5000) at room temperature. XPS was conducted using a Thermo Scientific (ESCALAB 250Xi). Optical micrographs were obtained using a DM2700M optical microscope (Leica). All electrical measurements were performed using a Keithley 4200 A semiconductor parameter analyzer and a probe station in an N$_2$ environment in a glove box at room temperature. Nyquist plots were collected using an electrochemical workstation (CHI760E) at room temperature in ambient conditions. Nominal stress–strain testing was measured using a tensile testing machine (CMT6103) with a loading rate of 50 mm min$^{-1}$.

### Data availability

The data that support the plots in this paper and other findings of this study are available from the corresponding authors upon request. Source data are provided with this paper.

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

## Acknowledgements

This work was supported in part by the National Science Fund for Distinguished Young Scholars of China (T2125005) (W.X.),  the National Key R&D Program of China (2022YFE0198200, 2022YFA1204500, 2022YFA1204504) (W.X.), Tianjin Science Foundation for Distinguished Young Scholars (Grant No. 19JCJQJC61000) (W.X.), the Shenzhen Science and Technology Project (JCYJ20210324121002008) (W.X.), the Inter-Governmental International Scientific and Technological Innovation Cooperation Key Projects (SQ2021YFE011099) (W.X.), and the National Natural Science Foundation of China (62201290) (C.J.).

## Author contributions

W.X. conceived the work. Y.N. and W.X. constructed the research frame. Y.N. and H.H. prepared the devices. Y.N. performed the data measurements and did the simulation. Y.N., J.L. and Q.Y. designed the hardware system. Y.N., J.L., H.H., L.Y. and W.X. analyzed the experimental data and simulation results. Z.X., C.J. and L.L. helped improve data visualization. W.X. directed the project. All authors discussed the results and implications and commented on the manuscript at all stages.

## Competing interests

The authors declare no competing interests.
