## [Peer Review File · Nature Communications]

REVIEWER COMMENTS

Reviewer #1 (Remarks to the Author):

The work demonstrates the use of electrochromic neurotrophic transistors (ENT) for visualized in-sensor computing. The authors use an electrochromic neuromorphic transistor (ENT) as a representation of synaptic weight to mimic a "visualized bionic reflex system". The device program and signal processing the authors demonstrates is promising for image recognition applications, and the work highlights the need for developing new neuromorphic devices based on color change. Overall, this work is impressive for pointing out a new direction for the development of artificial synapse. Although some details that need to be further clarified, the manuscript is worth accepting.

1. In Figure 1g, the authors claimed to have verified the ion accumulation state in relation to the bias using Kelvin probe force microscopy. While the experimental results support the author's claims, the reviewer suggests that the experimental method used is relatively lacking and the author should provide more detailed description about the experiment. Because the structure and size of the test device in Kelvin probe force microscopy can influence the results.

2. What is the specific cations in the "ionic polymer-metal composite" in artificial whisker? Is it the EMIM+ mentioned in Figure 1? The mechanism of artificial whisker work needs to be explained in more detail.

3. The authors mentioned that the device boasted a swift reset time, with conductivity reverting to 1/3000th of its initial value in just around 1 s. Is this characteristic related to the introduction of the Nafion film? If so, please provide more explanations and data, e.g., data for devices without the Nafion film.

4. In Figure S18, 6.37 Hz of f_c may not be a representative value for high frequencies. It is recommended to replace the term "high-pass filtering characteristics" with "low-frequency suppression characteristics".

5. The author demonstrated a flexible artificial synapse device, which showed excellent electrical performance from the results. However, for flexible devices, mechanical stability is also important. What is the tensile strength of the device? What is the elongation at break?

Reviewer #2 (Remarks to the Author):

The manuscript presents an innovative approach in the design and implementation of an Electrochromic Neuromorphic Transistor (ENT) for visualized in-sensor computing. The work introduces a pioneering concept by integrating color weight updates as a measure of synaptic weight, a significant departure from conventional systems relying solely on electrical conductivity changes. The research covers a broad spectrum of applications, from artificial neural systems to bionic reflex arcs, demonstrating the versatility and potential impact of the proposed technology. The manuscript is well-written and supported by comprehensive experimental data, providing a solid foundation for its findings.

Here are some specific comments and suggestions:

1. The manuscript is comprehensive, but it could benefit from improved organization and clarity, especially in the Introduction and conclusion sections.
2. While the manuscript discusses the novelty of the proposed artificial neural systems leveraging ion conduction in various applications, a more explicit comparison with existing literature or similar technologies could enhance the discussion, particularly those incorporating ion conduction.
3. In the supplementary material video, what does each color represent for each line? To enhance clarity, it is suggested that the authors include appropriate explanations in the video to elucidate the entire process of change.
4. Provide more details on how the ENT array demonstrates recognition capability. Explain the role of the 9-channel input layer and how numerical weights are mapped onto conductance states. Clarify how the real-time changes in synaptic weights are visualized during the training process. Additionally, elaborate on the neural network architecture and its adaptability.

In summary, the manuscript presents a concept in future neuromorphic computing, and with some interesting demonstrations in application, it has the potential to make a significant contribution to the field. Addressing the specific points mentioned above will strengthen the manuscript for potential publication in Nature Communications.

Response to Reviewers' Comments

Many thanks for the reviewers' valuable comments and constructive suggestions to our manuscript. Revision has been made point-to-point to comply with the reviewer's remarks as follows. The remarks of the reviewers are in black, our responses are in blue, and the revised parts were marked in red color in revised manuscript.

Reviewer #1 (Remarks to the Author):

The work demonstrates the use of electrochromic neurotrophic transistors (ENT) for visualized in-sensor computing. The authors use an electrochromic neuromorphic transistor (ENT) as a representation of synaptic weight to mimic a "visualized bionic reflex system". The device program and signal processing the authors demonstrates is promising for image recognition applications, and the work highlights the need for developing new neuromorphic devices based on color change. Overall, this work is impressive for pointing out a new direction for the development of artificial synapse. Although some details that need to be further clarified, the manuscript is worth accepting.

1. In Figure 1g, the authors claimed to have verified the ion accumulation state in relation to the bias using Kelvin probe force microscopy. While the experimental results support the author's claims, the reviewer suggests that the experimental method used is relatively lacking and the author should provide more detailed description about the experiment. Because the structure and size of the test device in Kelvin probe force microscopy can influence the results.

RESPONSE :

Thank you for the patient review and constructive comment. We provided more detailed description about the experiment in the revised manuscript: The tested device has a structure consisting of layers labeled as ion gel, Nafion, and P3HT from bottom to top. The testing method utilized is the Kelvin Probe Force Microscopy (KPFM) mode, performed with a Bruker AFM system (SCM-PIT-V2; materials: 0.01-0.025 Ω -cm

Antimony (n) doped Si; frequency: 75 kHz; spring constant: 3 N m⁻¹; diameter of tip: 50 nm), with a scan rate of 1 Hz (resolution: 256 pixels × 256 pixels).

In the manuscript, we made detailed revisions.

In Fig 1g:

Fig. 1 | Design of an electrochromic neuromorphic transistor. g Corresponding CPD of a diode device of architecture glass/ITO/ion gel/NML/P3HT using the KPFM mode after top bias of 0 V (initial state), -2, -4, 2 V and 4 V on the AFM tip.

Revised parts in the Supplementary Information:

In page 2, line 28:

To investigate the surface potential, the Bruker AFM system was switched in situ to a KPFM mode using various tip voltages (SCM-PIT-V2; materials: 0.01-0.025 Ω-cm Antimony (n)-doped Si; frequency: 75 kHz; spring constant: 3 N m⁻¹; diameter of tip: 50 nm), at a scan rate of 1 Hz with a resolution of 256 pixels × 256 pixels.

2. What is the specific cations in the "ionic polymer-metal composite" in artificial whisker? Is it the EMIM⁺ mentioned in Figure 1? The mechanism of artificial whisker work needs to be explained in more detail.

RESPONSE :

Thank you for your valuable and instructive comment. The cations referred to here are not the ions (e.g., EMIM⁺) in the ion gel, but rather hydrated sodium ions. The artificial whisker consists of a Nafion membrane with a thickness of 117 μm and silver electrodes on both sides. Its actuation mechanism is illustrated in Figure R1²⁹: Hydrated

sodium ions are formed due to the alkaline washing step during the Nafion fabrication process. When a voltage bias is applied to the anode, the hydrated sodium ions migrate towards the cathode, resulting in the expansion of the whisker's end and deflection towards the anode.

Fig. R1 | The mechanism of the bending motion of the ionic polymer-metal composite actuator caused by cation migration²⁹.

In the revised manuscript, we included comprehensive explanations and elaborations.

In page 13, line 14:

For bionic actuation, we incorporated an ionic polymer-metal composite into the artificial whisker (Figure 4c, detailed description in Supplementary Methods)²⁹. The creation of hydrated sodium ions is pivotal during the alkaline washing step in the fabrication process to ensure the functionality of the artificial whisker. Specifically, within the main body of the artificial whisker, the alkaline washing step induces the formation of hydrated sodium ions. When subjected to an electric field, the whisker's end undergoes expansion and deflection towards the anode, propelled by the migration of hydrated sodium ions from the anode to the cathode.

In References, we added:

29 Zhang, H. *et al.* Low-voltage driven ionic polymer-metal composite actuators: structures, materials, and applications. *Adv. Sci.* **10**, 2206135 (2023).

3. The authors mentioned that the device boasted a swift reset time, with conductivity reverting to 1/3000th of its initial value in just around 1 s. Is this characteristic related to the introduction of the Nafion film? If so, please provide more explanations and data,

e.g., data for devices without the Nafion film.

RESPONSE :

Thank you for your helpful comment. The device demonstrated a rapid reset time, with conductivity reverting to 1/3000th of its initial value in just around 1 s. This is attributed to the inclusion of a Nafion layer, which blocks the injection of anions to the P3HT channel. Consequently, upon the removal of the spikes, the anions accumulated at the bottom of the ion gel swiftly return to their initial positions, leading to the rapid restoration of the system to its initial state.

To validate this finding, we fabricated synaptic transistors without the Nafion modification layer and applied the same 1.5-s spike (Supplementary Fig. 7). After a 40 s spike removal, the retained current still exceeded 50% of the initial value, indicating the Nafion layer's critical role in the device's STP. Additionally, we further compared the current decay after applying and removing 100 spikes of -4 V for devices with and without Nafion (Supplementary Fig. 12). The results provide further support for this conclusion.

In the revised manuscript, we provide more explanations and data.

In page 9, line 11:

Comparing the decay of EPSC gain (ΔEPSC) and ΔGray value against the reset time, after a 1.5-s spike, ΔEPSC declined to 1/3000 of its initial value in just ~ 1 s, concurrently with ΔGray returning to 0. This is because Nafion hinders the doping of anions into the P3HT channel. Consequently, the anions swiftly return to their initial positions after the electrical spikes are removed, leading to the rapid restoration of the system to its initial state (Supplementary Fig. 7).

In page 10, line 16:

The ENT's short reset time of ~ 1 s makes it highly suitable for a visualized pattern-recognition network without the need for actively executing additional weight-refreshing actions (Supplementary Fig. 12):

In Supplementary Information, we added Fig. S7 and Fig. S12:

Fig. S7 | SDDP of P3HT synaptic transistor without Nafion layer. EPSC of P3HT synaptic transistor without Nafion layer triggered by a spike with durations of 1.5 s.

Fig. S12 | PSC retention rates. PSC retention rates of two types of synaptic transistors after 100 spikes of -4 V.

4. In Figure S18, 6.37 Hz of f_c may not be a representative value for high frequencies. It is recommended to replace the term “high-pass filtering characteristics” with “low-frequency suppression characteristics”.

RESPONSE :

Thank you for the valuable and instructive comment. To ensure a more precise description, we replaced the term "high-pass filtering" with "low-frequency suppression" in the manuscript.

Revised parts in the manuscript:

In page 12, line 1:

Moreover, synapses with a low probability of neurotransmitter release exhibit a

low-frequency suppression filtering effect: allowing the passage of signals with frequencies exceeding a cut-off value (f_c), while suppressing signals below this threshold²⁸. Our ENT exhibited similar **low-frequency suppression** filtering behavior, responding to stimuli with varying frequencies, exemplified by inputting a city image to simulate dynamic filtering functions (Supplementary Figs. 17-20, detailed in Supplementary Methods and Supplementary Text 4). These patterns could encode large-scale real-time dynamic signals.

Revised parts in the Supplementary Information:

In page 3, line 15:

Image processing and **low-frequency suppression** filtering simulation.

Images were extracted and processed using Python. The RGB was reduced to grayscale as $\text{gray} = \text{red} \times 0.299 + \text{green} \times 0.587 + \text{blue} \times 0.114$. To simulate a **low-frequency suppression** filter, a grayscale image of a city as an example was first converted from spatial domain to frequency domain by Fourier transform, then the matrix in the frequency domain was rearranged by moving the zero-frequency component to the center. An $m \times n$ filter template was proposed depending on the size of the generating new matrix. A **low-frequency suppression** filter was established by exploiting the filtering behavior of the ENT **low-frequency suppression** filter.

In page 7, line 9:

Supplementary Text 4. **Low-frequency suppression** filtering behavior

To quantify the **low-frequency suppression** filtering behavior of ENT devices, the EPSC was characterized in response to various stimuli that had a range of different f (Supplementary Fig. 17).

In Fig. 21:

Fig. S21 | SFDP index. Dependence of the gain of EPSC (A_{10}/A_1) on the pulse frequency; inset: schematic diagram of a **low-frequency suppression** filtering function.

In Fig. 22:

Fig. S22 | Low-frequency suppression filtering simulation. The original image of a city (left), and the image of the city after sharpening with the **low-frequency suppression** filtering function at the cut-off frequency of 6.37 Hz (right).

5. The author demonstrated a flexible artificial synapse device, which showed excellent electrical performance from the results. However, for flexible devices, mechanical stability is also important. What is the tensile strength of the device? What is the elongation at break?

RESPONSE :

Thanks for your valuable comment. To verify the mechanical stability of the

device, we conducted additional tests related to nominal stress-strain. The results showed that the device could withstand a maximum force of 391.72 N, and its maximum tensile strength reached 217.62 MPa. At the point of fracture, the elongation reached 73.77%. These results demonstrate the device's excellent mechanical flexibility and stability.

In the revised manuscript, we added more experimental data and analysis.

In page 12, line 29:

To access the flexibility of ENT during bio-integration, a nominal stress-strain test was conducted (Supplementary Fig. 26). As the tensile strain increased, the ENT exhibited successive stages including the elastic stage, plastic yield stage, and necking stage before eventual fracture. At the point of fracture, the elongation reached 73.77%. The ENT demonstrated a maximum force tolerance of 391.72 N and a maximum tensile strength of 217.62 MPa, meeting the necessary criteria for bio-integration.

Revised parts in the Supplementary Information:

In page 3, line 12, we added:

Nominal stress-strain testing was measured using a tensile testing machine (CMT6103) with a loading rate of 50 mm/min.

We added Fig. S26:

Fig. S26 | Nominal stress-strain characteristic. **a** Nominal stress-strain testing diagram. **b** Nominal stress-strain curve of ENT.

Reviewer #2 (Remarks to the Author):

The manuscript presents an innovative approach in the design and implementation of an Electrochromic Neuromorphic Transistor (ENT) for visualized in-sensor computing. The work introduces a pioneering concept by integrating color weight updates as a measure of synaptic weight, a significant departure from conventional systems relying solely on electrical conductivity changes. The research covers a broad spectrum of applications, from artificial neural systems to bionic reflex arcs, demonstrating the versatility and potential impact of the proposed technology. The manuscript is well-written and supported by comprehensive experimental data, providing a solid foundation for its findings.

In summary, the manuscript presents a concept in future neuromorphic computing, and with some interesting demonstrations in application, it has the potential to make a significant contribution to the field. Addressing the specific points mentioned above will strengthen the manuscript for potential publication in Nature Communication.

Here are some specific comments and suggestions:

1. The manuscript is comprehensive, but it could benefit from improved organization and clarity, especially in the Introduction and conclusion sections.

RESPONSE :

We appreciate your suggestion regarding the organization and clarity of the manuscript, particularly in the Introduction and conclusion sections. We made significant revisions to improve the overall structure and coherence of the manuscript. We believe that these revisions have significantly enhanced the organization and clarity of the manuscript, aligning it more closely with the standards expected by reviewers.

In page 4, line 14:

The communication of color information stands as one of the most immediate and widespread methods of interaction among biological entities¹¹; **it indicates that the ability to integrate color information into artificial neural systems can offer numerous advantages and opportunities for enhancing their functionality.** For instance, chameleons can adjust the activity of pigment cells in response to their surroundings by

modulating the concentration of catecholamine neurotransmitters like epinephrine (EPI), enabling rapid color changes for communication with conspecific individuals (Figure 1a, top)¹². Despite the direct and efficient nature of color weight updates for information exchange, there has been a notable scarcity of neuromorphic electronics capable of utilizing color changes to convey information.

In most organisms, the neuro-reflex process adheres to an all-or-none law, where an action is triggered only when the stimulus intensity surpasses a certain threshold¹³⁻¹⁵. **Most existing neuromorphic electronics focus primarily on pure electrical signal processing and do not fully exploit the potential of incorporating color-based information; such reflex systems rely solely on electrical signal transmission might inadvertently filter out certain weak yet potentially significant signals¹⁶⁻¹⁸.** By combining color-based alterations with adaptable electrical properties, it becomes possible to visualize changes in synaptic weight and monitor a broader range of environmental signals compared to conventional bionic reflex systems. **Moreover, the integration of color information as an additional measure of synaptic weight in visualized artificial neural systems can enhance the accuracy and efficiency of in-sensor computing tasks. This approach enables more nuanced processing capabilities, mirroring the complexity and richness observed in natural sensory systems.**

In page 14, line 18:

It showcases visualized weight updating, simulating the multi-dimensional tuning of synaptic weight by adjusting both neural impulse information weight and pigment cell chromaticity information weight. Leveraging a unique mechanism that adaptively screens ions through the NML, the device also achieved rapid reconfigurable electrical characteristics. Notably, the process exhibited a remarkably swift recovery time of less than 1 s, facilitating rapid signal transmission without interference. Capitalizing on these inherent features, we demonstrated an electro-chromaticity boost coding approach and developed a strategy for constructing a visualized pattern-recognition network. Moreover, the integration of the ENT with an artificial whisker led to the creation of a visualized neural reflex mirroring that of the longicorn beetle. This achievement marked the first visualization of signal flow, effectively enabling the monitoring of

minute environmental stimuli during reflex actuation. Our work has the potential to enhance the functionality and adaptability of artificial neural systems, thereby paving the way for the development of more sophisticated and bio-inspired computing systems.

2. While the manuscript discusses the novelty of the proposed artificial neural systems leveraging ion conduction in various applications, a more explicit comparison with existing literature or similar technologies could enhance the discussion, particularly those incorporating ion conduction.

Response:

We thank the reviewer for the valuable comments. In order to emphasize the significance of our work, in the revised manuscript, we conducted a comparative analysis of recent works related to artificial neural systems leveraging ion conduction in terms of biomimicking concept, key processing unit, perception type, and response performance (Table S2). Our visualized bionic reflex system represents a pioneering approach by integrating motor responses with color information, providing inspiration for related biomimetic systems.

In the revised manuscript, we added more discussions.

In page 14, line 10:

The integration of motor responses with color information in this concept contributes to proposing new paradigms for artificial neural systems that leverage ion conduction^{14,18,30-35}.

In Supplementary Information, we added Table S2:

Table S2 Comparison of recent works relating to artificial neural systems leveraging ion conduction.

Biomimicking concept	Key processing unit leveraging ion conduction	Perception type	Response performance		Year	Ref.
			Actuation	Chrominance-change		
Biological motor system	Optical synaptic device	Photoelectric signal	Strain: 35%	Not given	2019	30

Neurologically integrated soft engineering systems	Rubbery synaptic transistor	Piezoelectric signal	Curvature: 30°	Not given	2019	31
Artificial somatic reflex arc	Threshold controlling unit	Piezoelectric signal	Reversing actuation	Not given	2020	14
Synaptic device amplifier circuit-polymer actuator system	Graphdiyne-based artificial synapse	Electrical signal	Curvature: 20°	Not given	2021	32
High-Strength Neuromuscular System	Parallel-channeled artificial synapse	Electrical signal	Force: 29.2 N; Curvature: 73°	Not given	2022	33
Artificial visual nerve	Perovskite synaptic device	Photoelectric signal	Force: 21.6 N	Not given	2022	34
Low-power stretchable neuromorphic nerve	Stretchable synaptic transistor	Electrophysiological signals	Force: 412 mN Curvature: 40°	Not given	2023	18
Artificially-intelligent cornea	ZTO-fibers artificial synapse	Photoelectric signal	Not given	From light blue to dark blue	2023	35
Visualized bionic reflex system	Electrochromic neuromorphic transistor	Piezoelectric signal	Curvature: 93°	Gray-change: 20		Our work

In References:

- 14 He, K. *et al.* An artificial somatic reflex arc. *Adv. Mater.* **32**, e1905399 (2020).
- 18 Lee, Y. *et al.* A low-power stretchable neuromorphic nerve with proprioceptive feedback. *Nat. Biomed. Eng.* (2022).
- 30 Karbalaee Akbari, M. & Zhuiykov, S. A. bioinspired optoelectronically engineered artificial neurobotics device with sensorimotor functionalities. *Nat. Commun.* **10**, 3873 (2019).
- 31 Shim, H. *et al.* Stretchable elastic synaptic transistors for neurologically integrated soft engineering systems. *Sci. Adv.* **5**, eaax4961 (2019).
- 32 Wei, H. *et al.* Mimicking efferent nerves using a graphdiyne-based artificial synapse with multiple ion diffusion dynamics. *Nat. Commun.* **12**, 1068 (2021).

- 33 Ni, Y. *et al.* A high-strength neuromuscular system that implements reflexes as controlled by a multi-quadrant artificial efferent nerve. *ACS Nano*. **16**, 20294-20304 (2022).
- 34 Gong, J. *et al.* An artificial visual nerve for mimicking pupil reflex. *Matter*. **5**, 1578-1589 (2022).
- 35 Qu, S. *et al.* An artificially-intelligent cornea with tactile sensation enables sensory expansion and interaction. *Nat. Commun.* **14**, 7181 (2023).

3. In the supplementary material video, what does each color represent for each line? To enhance clarity, it is suggested that the authors include appropriate explanations in the video to elucidate the entire process of change.

RESPONSE :

Thank you for your helpful comment. In the supplementary material video, the three lines (Right figure) represent the curves of the RGB primary colors' variations, which are extracted from the real color change observed in the sampling region (Left figure) during Morse code encoding.

In the revised video, we marked appropriate explanations to elucidate the entire process of change.

4. Provide more details on how the ENT array demonstrates recognition capability. Explain the role of the 9-channel input layer and how numerical weights are mapped onto conductance states. Clarify how the real-time changes in synaptic weights are visualized during the training process. Additionally, elaborate on the neural network architecture and its adaptability.

RESPONSE :

Thanks for the valuable comment. In the revised manuscript, we provided further elaboration on how the ENT array demonstrates recognition capability. Additionally, we explained the role of the 9-channel input layer and how numerical weights are mapped onto conductance states: The 9-channel input layer, represented by different V_{DS} , reflects pixel information. V_{DS} acts as the reading voltage, controlled by the extracted eigenvectors from classified samples, simulating the reception of visual information. The conductivity state is adjusted by varying the number of pulses applied

to the device within a short period, assigning different weights to the received visual information. The current programming is determined by the product of the reading voltage signal and conductance state, executed through the crossbar to output pre-processed results.

To clarify how real-time changes in synaptic weights are visualized during the training process, we supplemented the chromaticity-current map of the ENT device triggered by spikes of -4 V as a colorimetric card that holds the chromaticity information of red-green-blue (RGB) and fixed gray (Supplementary Fig. 14). Furthermore, we explained that the real-time changes in visualized synaptic weights can be retrieved by referring to the colorimetric card.

For the neural network architecture and its adaptability, we provided more comparisons regarding plasticity (Supplementary Fig. 12), and expounded that the ENT's short reset time of ~ 1 s makes it well-suited for a visualized pattern-recognition network without actively performing additional weight refreshing actions. Therefore, when applied to vision-related neural network architecture, the device unit enhances computational efficiency and reduces energy consumption in the central brain by pre-processing signals at the perceptual end.

In the revised manuscript, we added more experimental data and analysis.

In page 10, line 16:

The ENT's short reset time of ~ 1 s makes it highly suitable for a visualized pattern-recognition network without the need for actively executing additional weight-refreshing actions (Supplementary Fig. 12): a 'neural core' with a high refresh rate, enabling accelerated matrix multiplication achieved by applying various V_{DS} amplitudes (-0.1 V to -0.5 V) and different spike numbers (1 to 100) (Figure 3c).

In page 10, line 25:

This pre-processing of signals at the perceptual end enhances computational efficiency and reduces energy consumption in the central brain. Drawing inspiration from the efficacy and transient nature of visual memory, we designed a 3×3 ENT array to mimic the visual cortex (Figure 3d). Each ENT functions within a crossbar array, operating in a voltage-controlled capacity in the circuit schematic, performing matrix multiplication

and outer-product update operations: V_{DS} serves as the reading voltage controlled by the extracted eigenvectors from classified samples simulating the reception of visual information. The conductivity state is adjusted by varying the number of pulses applied to the device within a short period, assigning different weights to the received visual information; the current programming is determined by the product of the reading voltage signal and conductance state, executed through the crossbar to output pre-processed results.

In page 11, line 20:

Additionally, real-time changes in synaptic weights can be visualized during the training process (Figure 3h; Supplementary Fig. 14): we can retrieve the EPSC mapping value by referring to a colorimetric card that holds the chromaticity information of red-green-blue (RGB) and fixed gray.

In Supplementary Information, we added Fig. S12 and Fig. S14:

Fig. S12 | PSC retention rates. PSC retention rates of two types of synaptic transistors after 100 spikes of -4 V.

Fig. S14 | Chromaticity-current map. Chromaticity-current (EPSC) map of the ENT device triggered by spikes of -4 V.

REVIEWERS' COMMENTS

Reviewer #1 (Remarks to the Author):

The authors fully addressed the comments raised by the reviewers. The scientific quality is dramatically improved. Therefore, this manuscript can be published in Nature Communications without further revisions.

Reviewer #2 (Remarks to the Author):

The authors have addressed most of technical questions raised by the reviewers. In the introduction part, the authors are suggested to cite early works on "in-sensor computing", such that the readers can understand the context. I would recommend the acceptance.

Response to Reviewers' Comments

Many thanks for the reviewers' valuable comments and constructive suggestions to our manuscript. Revision has been made point-to-point to comply with the reviewer's remarks as follows. The remarks of the reviewers are in black, our responses are in blue, and the revised parts were marked in red color in revised manuscript.

Reviewer #1 (Remarks to the Author):

The authors fully addressed the comments raised by the reviewers. The scientific quality is dramatically improved. Therefore, this manuscript can be published in Nature Communications without further revisions.

Reviewer #2 (Remarks to the Author):

The authors have addressed most of technical questions raised by the reviewers. In the introduction part, the authors are suggested to cite early works on "in-sensor computing", such that the readers can understand the context. I would recommend the acceptance.

RESPONSE:

Thank you for the valuable and instructive comment. In the revised manuscript, we included references to early works on "in-sensor computing" in the introduction section to provide better context and understanding for the readers.

We are grateful for your recommendation to accept the manuscript, and we hope that our revisions have addressed any remaining concerns.

In page 4, line 4:

Moreover, the integration of color information as an additional measure of synaptic weight in visualized artificial neural systems can enhance the accuracy and efficiency of in-sensor computing tasks by high-fidelity processing data locally¹⁹⁻²¹.

In References, we added:

- 19 Chai, Y. In-sensor computing for machine vision. *Nature* **5**, 579 (2020).
- 20 Zhou, F., & Chai, Y. Near-sensor and in-sensor computing. *Nat. Electron.* **3**, 664-671 (2022).
- 21 Wan, T., Shao B., Ma S., Zhou Y., Li Q., & Chai Y. In-sensor computing: materials, devices, and integration technologies. *Adv. Mater.* **35**, 2203830 (2023).